# S1P/S1P Receptor Signaling in Neuromuscolar Disorders

**DOI:** 10.3390/ijms20246364

**Published:** 2019-12-17

**Authors:** Elisabetta Meacci, Mercedes Garcia-Gil

**Affiliations:** 1Clinical Biochemistry and Clinical Molecular Biology Unit, Department of Experimental and Clinical Biomedical Sciences “Mario Serio”, University of Florence, Viale Pieraccini 6, 50139 Florence, Italy; 2Interuniversity Institute of Myology, University of Firenze, 50134 Firenze, Italy; 3Department of Biology, Unit of Physiology, University of Pisa, via S. Zeno 31, 56127 Pisa, Italy; mercedes.garcia@unipi.it; 4Interdepartmental Research Center “Nutraceuticals and Food for Health”, University of Pisa, 56127 Pisa, Italy

**Keywords:** sphingolipids, sphingosine 1-phosphate receptors, ceramide, skeletal muscle, nervous system, neuromuscular disease, Charcot-Marie-Tooth disease, myasthenia gravis, duchenne muscular dystrophy

## Abstract

The bioactive sphingolipid metabolite, sphingosine 1-phosphate (S1P), and the signaling pathways triggered by its binding to specific G protein-coupled receptors play a critical regulatory role in many pathophysiological processes, including skeletal muscle and nervous system degeneration. The signaling transduced by S1P binding appears to be much more complex than previously thought, with important implications for clinical applications and for personalized medicine. In particular, the understanding of S1P/S1P receptor signaling functions in specific compartmentalized locations of the cell is worthy of being better investigated, because in various circumstances it might be crucial for the development or/and the progression of neuromuscular diseases, such as Charcot–Marie–Tooth disease, myasthenia gravis, and Duchenne muscular dystrophy.

## 1. Introduction

The bioactive sphingolipid metabolite, sphingosine 1-phosphate (S1P), and the signaling pathways triggered by its binding to specific G protein-coupled receptors (S1PRs) play a critical regulatory role in many pathophysiological processes, including skeletal muscle and nervous system degeneration. Sphingolipids (SLs) are involved in the process of muscle fatigue delay [1,2,3] and in the potentiation of skeletal tissue strength, mass, and metabolism [4,5].

S1P acts as a skeletal muscle trophic factor regulating satellite cell proliferation and tissue repair [6,7,8]. The S1P/S1PR signaling has been demonstrated to play a crucial role in terminal myogenic differentiation [9,10,11,12,13]. In particular, the receptor subtypes S1PR2 and S1PR3 have opposite roles in cell proliferation [4,5,6,7,8,9,10,11,12,13,14] and the bioactive lipid promotes stem cell progression by repressing cell cycle inhibitors via S1PR2/ Signal transducer and activator of transcription 3-dependent signaling [4]. Overexpression of S1PR3 suppresses cell cycle progression, while satellite cells isolated from S1PR3-null mice exhibit enhanced proliferation [14]. S1PR3 has been demonstrated to be a target of miR-127, whose action is required for myogenic cell differentiation [15]. Recent findings support the role of the S1P/S1PR axis also in the cell mature differentiated phenotype. Indeed, down-regulation of sphingosine kinase 1 (SphK1) activity, increased expression of the transport spinster homolog 2 (SPNS2), and a shift in the expression of S1PR2 and S1PR1/S1PR3 have been demonstrated in myotubes induced to atrophy by dexamethasone treatment, thus, indicating a positive role of S1P/S1PR signaling in the maintenance of the normal terminally differentiated myotube phenotype [16]. 

Several studies have highlighted the signaling pathways activated by S1PRs in skeletal muscle cells. Exogenous S1P induces a rapid stimulation of phospholipase D activity [17] and a transient and rapid membrane association of RhoA protein [18] through the activation of protein kinase C in C2C12 cells. S1P/S1PR axis is also involved in Ca^2+^ mobilization [19] and cytoskeletal remodeling. S1P induces stress fibers formation and this event is associated with plasma membrane tension and activation of mechanosensitive channels, namely stretch-activated channels, which play an essential role in the early proliferative stage of myoblasts as well as in myogenic differentiation [20,21,22,23,24]. S1P/S1PR signaling is also able to regulate extracellular matrix remodeling and metalloprotease activity/expression in skeletal muscle cells [5,25,26].

S1P regulates cell survival, apoptosis, autophagy, and differentiation and migration of different cells of the central nervous system (neurons, astrocytes, oligodendrocytes, and microglia) [27,28,29]. S1P also modulates inflammation, and excitability [28] and it has a crucial role in brain development. In fact, SphK1/SphK2-double knockout mice [30] and S1P1R knockout mice show impaired neurogenesis (decreased proliferation and increased apoptosis) and embryonic death. Modifications of S1P metabolism, S1PR expression profile, and S1P-mediated signaling have been described in neurodegenerative diseases, including multiple sclerosis, Alzheimer’s disease, Parkinson’s disease, Huntington’s disease for recent reviews see [29,31,32] and neuropathic pain [33,34,35,36,37]. Some S1P analogues have been approved for the treatment of multiple sclerosis or have passed phase III trial [31,38], while the effects of other enzyme activators and inhibitors and S1PR agonists/antagonists are being tested in animal models of neurodegenerative diseases. For example, an activator of SphK1 and an agonist S1PR5 exert beneficial effects on a mouse model of Huntington’s disease [39,40].

This review focuses on the current knowledge of how the localized actions of S1P and/or S1PR signaling affect different biological processes in skeletal muscle and nervous system. In particular, the attention will be paid on the role of S1P/S1PR signaling in the control of mitochondrial and nuclear functions, that are much more complex than previously thought. Part of this review will also be dedicated to the involvement of S1P and S1PR-mediated signaling as potential druggable targets in neuromuscular diseases, such as Charcot-Marie-Tooth disease (CMT), myasthenia gravis (MG) and Duchenne muscular dystrophy. 

## 2. Sphingolipid Metabolism

Sphingolipids (SLs) are a large group of sphingoid metabolites which include signaling molecules associated with cellular activities that are crucial in physiological and pathological conditions [41,42]. In fact, SLs represent the second most abundant structural lipids in the cellular membrane and a network of bioactive molecules with relevant biological functions. SLs are made by sphingosine (Sph) backbone linked to one hydrophobic acyl chain and a phosphate head group ester. The more abundant SLs in the plasma membrane of mammalian cells (reviewed in [43]) is sphingomyelin (SM), which possesses a phosphorylcholine headgroup associated to the sphingoid base [42]. SM breakdown begins from its hydrolysis by sphingomyelinases, SM specific type-C phospholipases generating phosphocholine released into the aqueous environment and ceramide (Cer) that diffuses through the membrane. The de novo synthesis of SLs starts from serine and palmitate condensation catalized by serine palmitoyl transferase [44]. The product, 3-keto-dihydrosphingosine, is then reduced to sphinganine via 3-ketosphinganine reductase and subsequently acylated by (dihydro)-ceramide synthases [45,46] to form dihydroceramide which is subsequently reduced to Cer by dihydroceramide desaturase. De novo Cer synthesis occurs in the endoplasmic reticulum with Cer being transported from the endoplasmic reticulum to the Golgi apparatus by either vesicular trafficking or by the Cer transfer protein [46]. Cer can be also phosphorylated to ceramide 1-phosphate (C1P) and is the substrate for ceramidase which cleaves fatty acids from Cer producing Sph. Subsequent phosphorylation of Sph by two distinct SphK isoforms (SphK1 and SphK2) [47,48] produces the bioactive lipid S1P. Sphingosine-1-phosphate phosphatase (SPP) causes S1P dephosphorylation to Sph, whereas S1P lyase (S1PL) degrades it to hexadecenal and ethanolamine phosphate [49]. 

It is worth to note that the interplay and the ratio between S1P and its metabolic precursors, in particular Cer and Sph, is a crucial determinant of cellular fate [50,51]. In fact, Cer and Sph can regulate growth arrest, senescence and apoptosis, whereas S1P and C1P control cell proliferation and cell survival. An interesting feature, worthy of remark, is the distinct intracellular compartmentalization of the biosynthesis of some SLs. Therefore, the activity of specific enzymes produces distinct pools of SLs in different organelles, likely allowing specific functions.

## 3. S1P/S1PR Signaling

Because of its peculiar mechanism of action, S1P is the most relevant molecule among the bioactive SLs [52,53]. Indeed, S1P can act as an intracellular mediator as well as a ligand of specific heterotrimeric GTP binding protein-coupled receptors, initially named endothelial differentiation genes (EDG)1-5, and at present *S1PR1-5* [54,55]. 

Some intracellular targets of S1P are documented (reviewed in [29,52]) (Figure 1). One of them is prohibitin 2 (PHB2), a highly conserved protein that regulates mitochondrial assembly and functions [56]. Another is the tumor necrosis factor receptor-associated factor 2 (TRAF-2), a key adaptor molecule in tumor necrosis factor receptor signaling complexes, which has an E3 ubiquitin ligase activity and is a key component of the nuclear factor-κB (NF-κB) pathway [57], crucially involved in inflammatory gene regulation [58].

In addition, recent studies suggest the binding of S1P to the histone deacetylases 1/ 2 (HDACs) [59], human telomerase [60], PARPγ [61] and atypical protein kinase C [62]. Moreover, S1P specifically interacts with the N-terminal domain of the heat shock proteins GRP94 and HSP90α [63].

Although many functions have been attributed to S1P, some of its effects remain unexplored and are likely mediated by unknown intracellular targets. Notably, an intracellular action of S1P via intranuclear localised S1PRs cannot be ruled out. Indeed, S1PR5 has been found in centrosomes [64] and S1PR2 translocates to the nucleus in breast cancer cells [65]. Evidence obtained from immunohistochemistry also suggests nuclear localizations in different human tissues [66].

Several years ago, Spiegel and collaborators proposed the paradigm of “inside-out” signaling: once synthesized inside the cell, S1P can be released out of cells and act as an autocrine or paracrine signal. Since S1P is relatively hydrophilic due to its charged polar head group, it is unable to diffuse over the membrane and requires transporters to exit the cell including ATP-binding cassette transporters, major facilitator superfamily transporter 2b, and the SPNS2 [67,68]. Although ABC transporters were originally perceived as pore-forming proteins with an aqueous pore acting as a channel for hydrophilic substrates, they might function as a floppase, moving lipid soluble molecules from the inner to the outer plasma membrane leaflet [69].

SNPS2 is a member of non-ATP-dependent organic ion transporter family that plays a crucial role in the physiology of immune and vascular systems and, as reported recently also in tumor metastasis [67]. Released S1P can signal as an autocrine or paracrine molecule by binding to its specific S1PRs [59,70,71,72]. 

S1PRs are differently expressed in normal and malignant human tissues and mainly localized at plasma membrane [42,73]. Moreover S1PRs are coupled to one or more monomeric G proteins [26,42,54], which specify the downstream signaling targets of each receptor subtype, thus indicating that S1P action can be highly modulated and mediated by many signaling pathways. 

Two functional nuclear export signal sequences are responsible for SphK1 localization in the cytosol [74], whereas SphK2 has nuclear import and export sequences, and is found predominantly in the nucleus [75] as well as in mitochondria. Thereby, S1P formed by SphK1, which translocates after activation to the plasma membrane, mediates cell proliferation and survival [76]. This is confirmed by observations indicating that enhanced SphK1 activity, is strictly correlated with neoplastic transformation and tumorigenesis [77,78,79] and pharmacological inhibition of the enzyme attenuates tumor growth in numerous animal models [80]. 

Although both SphK isoenzymes catalyze the same reaction, many recent studies have found that SphK2 can promote cell cycle arrest and apoptosis, an opposite effect compared to SphK1. Moreover, when it is located into the nucleus, SphK2 mediates DNA synthesis inhibition and HDAC regulation [59,75] (see Section 4.2), whereas when it is present in the mitochondria, promotes programmed cell death by collaborating with pro-apoptotic proteins, such as Bax [81,82,83].

Interestingly, although for many years an oncogenic function has been attributed only to SphK1, recent studies have demonstrated a similar role for SphK2 in various solid and liquid tumors, such as lymphoblastic leukemia [84,85]. Moreover, it was found that in in vivo studies, the blockage of SphK2 signaling significantly reduces tumor growth of human xenograft models in mice [86], and enhanced enzyme expression is associated with non-small cell lung cancer and multiple myeloma [72,87]. In addition, the silencing of SphK2 expression induces cell death and increases sensibility to various cancer cell types [88,89].

All these reports underline the relevance of both SphK isoforms in mediating tumorigenesis. Regarding this feature, it is worth to note that despite the differences in cellular localization and regulators, these isoforms can compensate for each other as demonstrated in SphK1 and SphK2 knock-out mice. Indeed, the knock-outs of the single gene are viable, whereas the double knock-outs are embryonically lethal [30]. Moreover, SphK1 expression and S1P circulating level are increased in Sphk2-null mice, likely due to a compensatory effect of SphK1 [90]. 

The in vivo S1P level is higher in blood stream than in tissues [91,92,93]. This is due to the release of S1P from many blood cells and the lack of the enzymes involved in its catabolism in platelets and erythrocytes [94,95]. Several years ago it has been reported that this S1P gradient, formed between tissue and blood, is physiologically relevant in the control of immune cells when, in particular conditions (i.e., tissue damage), they egress from lymphoid organs to circulation [77,96]. Notably, the response to S1P gradients is strictly dependent on the distinct profile of S1PR subtypes expressed in different cells in specific circumstances. 

### 3.1. S1PR in Nervous System 

S1PRs are expressed in neurons, astrocytes, microglia, and oligodendrocytes (for a review see [97]) and they are involved in cell survival, proliferation, differentiation, migration, and inflammation [27,28,29]. S1PR expression pattern in the central nervous system depends not only on the cell type, but also on the developmental stage, central nervous system region and physiopathological conditions. For example, in astrocytes S1PR1 and S1PR3 are the most abundant receptor subtypes, but the expression of S1PR5 increases following exposure to growth factors [98]. Therefore, it may play a role in the differentiation of radial astrocytes during development [99]. In addition, expression of S1PR1 and S1PR3 is increased in reactive astrocytes in multiple sclerosis lesions and in cultured astrocytes under proinflammatory circumstances [100,101]. Recently, it has been demonstrated that S1PR3 mediates RhoA activation and induces transcription of cyclooxygenase-2, interleukin-6 (IL-6), and VEGFa mRNA in astrocytes [102]. 

In addition to its contribution to neuroinflammation, there is also evidence that S1P generated in the dorsal horn of the spinal cord in response to nerve injury leads to neuropathic pain by activating S1PR1 in astrocytes [103]. Microglia activation by lipopolysaccharide stimulation is associated with downregulation of S1PR1 and without any change in S1PR2 and S1PR3 [104]. Recent data indicate that S1PR1-3 are associated with M1 microglial polarization in the ischemic brain by regulating ERK1/2, JNK, p38 MAPKs, and Akt [105,106,107]. S1PR5 is preferentially expressed in oligodendrocytes [108]. S1PR5 activation modulates two distinct functional pathways mediating either the process retraction in immature oligodendrocytes as well as cell survival in mature cells [109]. 

Interestingly, S1PR2 binds at different sites both S1P and Nogo-A, a protein that inhibits neurite growth and synaptic plasticity [110]. Kempf et al. [110] have demonstrated that S1PR2 blockage enhances plasticity in hippocampus and motor cortex and have suggested that the binding of S1P and Nogo-A to the same receptor could facilitate the fine-tuning of cellular responses. S1PR1-3, and S1PR5 are expressed in rat hippocampal neural progenitor cells [111] and human embryonic stem cell-derived neural epithelial progenitors [112]. S1PR1 plays an important role in the transplanted neural stem cell migration toward injured area of spinal cord [113], while pharmacologic or genetic inhibition of S1P2R enhances S1P-mediated neural progenitor cell migration in the ischemic brain without affecting proliferation and differentiation [114]. S1PR1 increases neurogenesis (proliferation and differentiation of neural stem cells) in hippocampus after traumatic brain injury [115] improving memory and learning performance. 

Fingolimod (FTY720) is a structural analogue of Sph, which is phosphorylated in vivo by SphK2 and converted to fingolimod-phosphate, a structural analogue of S1P. The phosphorylated compound acts as an agonist of S1PR1, S1PR2, S1PR3, and S1PR5 and causes the irreversible internalization and degradation of bound receptors, thereby acting as functional antagonists [29,116]. Fingolimod (FTY720) promotes the proliferation of embryonic neural stem cells (NSCs), enhances hippocampal neurogenesis and learning and memory abilities in adult mice, probably through S1PR1 [117]. Treatment with fingolimod (FTY720) for seven weeks partially restores the pool of neural progenitor cells in irradiated mice and mitigates radiation-induced learning dysfunction [118]. But when mice are exposed to fingolimod (FTY720) prior to cranial irradiation, a neuroprotective effect without any change in neurogenesis has been reported [119]. Recently, it has been demonstrated that fingolimod (FTY720), at nanomolar concentrations, enhances the survival of NSCs and their differentiation into mature oligodendrocytes in vitro, primarily through S1PR3 and S1PR5 [120]. In vivo, treatment with either fingolimod (FTY720), or transplanted NSCs alone is ineffective, but the combination therapy of fingolimod (FTY720) and transplanted NSCs improves the chronic stage of experimental autoimmune encephalomyelitis, an animal model of multiple sclerosis [120]. More information about the role of SphK/S1PRs can be found in the recent review by Ng and collaborators [8]. Some sphingosine or S1P analogues have entered clinical studies for neuromuscular diseases, above all for multiple sclerosis and amyotrophic lateral sclerosis. Fingolimod (FTY720) is an immunosuppressant which prevents lymphocyte egress from the lymph nodes, thereby, reducing autoaggressive lymphocyte migration into the central nervous system. Fingolimod (FTY720) is currently used in the treatment of relapsing multiple sclerosis [121]. In addition, it is safe and well-tolerated and can reduce circulating lymphocytes in patients suffering from amyotrophic lateral sclerosis [122]. Other S1P analogues exhibiting higher S1PR specificity than fingolimod (FTY720) include siponimod/BAF13 (a potent agonist of S1PR1 and S1PR5 but not of S1PR3), and RPC1063/ozanimod have entered phase III clinical trials for multiple sclerosis [38,123,124].

### 3.2. S1PR in Skeletal Muscle System 

S1PR-dependent signaling has been shown in skeletal muscle cells. Mouse myogenic C2C12 cell line obtained from satellite cells express S1PR1, S1PR2, and S1PR3 at mRNA level and S1PR1 expression seems to be higher than the other subtypes [125]. Notably, during the myogenic processes the expression of S1PR2 progressively decreases, whereas S1PR3 becomes higher by the time the myotubes are terminally differentiated [125], supporting a different role in myoblast differentiation and terminally differentiated myotubes. In fact, it has recently been reported the crucial role of S1PRs in the maintenance of mature myotube phenotype: S1PR2 expression is reduced and S1PR3 increased in C2C12 myotubes induced to atrophy by glucocorticoid dexamethasone treatment [16]. The down-regulation of S1PR1 and S1PR3 also occurs after denervation confirming the trophic action of S1P/S1PR [126]. In an ex-vivo murine model of muscle damage induced by eccentric contraction, the bioactive lipid is able to reduce the deposition of collagen and promote the expression of metalloprotease-9, indicating the ability of S1P to affect the stiffness of the extracellular matrix and then satellite cell behavior [12,127]. In addition, Bondì et al. [128] observed that S1P/S1PR3 axis plays a negative role in muscle mass maintenance and force in soleus during aging and extracellular S1P exerts a protective effect during muscle fatigue development [129]. S1P/S1PR3 signaling also modulates excitation-contraction coupling and intracellular calcium mobilization [121,122,123,124,125,126,127,128,129,130]. 

During the regeneration process following injury the expression of S1PR1 increases, while S1PR3 expression, which is higher at the beginning of this recovery process, progressively decreases. In fact, S1P-induced increase of regenerating fiber growth is reduced by the activation of S1PR1 and by the inhibition of S1PR3, suggesting that the early phases of regeneration require inhibition of S1PR1 and the activation of S1PR3 [126]. These results are consistent with the enhanced acute muscle regeneration found in S1PR3-null mice and the less severe dystrophic phenotype found in mdx mice with genetic ablation of S1PR3 [14]. S1PR2 is absent in quiescent cells, but starts to be expressed at the early regenerating phases. Exogenous S1P stimulates the growth of regenerating myofibers, and an anti-S1P monoclonal antibody neutralizes this action, thus, further supporting the S1PR–mediated action [131].

Exercise also increases S1P and S1PR1 levels in rodents and S1P content in humans [132,133] and improves insulin sensitivity in the rat skeletal muscle tissue [134]. 

Some effectors of the S1P/S1PR signaling in skeletal muscle have been uncovered. The functional interaction between stress fibers and stretch-activated ion channel is required for S1P-induced myogenic differentiation of C2C12 myoblasts [22]. Indeed, both the disruption of actin cytoskeleton and the impairment of channel functionality hamper the myogenesis promoted by the bioactive lipid. Another downstream event promoted by S1P/S1PR axis in skeletal muscle cells is the expression of the gap junctional protein connexin-43 (Cx43) [10]. Interestigly, S1P induces the physical association of Cx43 with cytoskeletal proteins (F-actin and cortactin) and, thereby, cytoskeleton remodeling required for myogenesis. Cx43/cytoskeleton interaction during myoblast differentiation is also dependent on the activation of transient receptor potential channel 1 (TRPC1), a stretch-activated ion channel [23]. Although some features regarding the role of S1P/S1PR signaling have been described, further investigation is required to clarify the physiological role of S1P/S1PR axis in vivo. 

## 4. Intracellular Action of S1P/S1PR Signaling

The plasma membrane as well as the internal membranes are scaffolds and inducers of intracellular signals. Emerging evidences underline the role of the SL metabolism in selected organelles, including mitochondria and nucleus, supporting compartmentalized functions for these bioactive molecules. Lipid microdomains, with specific lipid composition and metabolic/signaling proteins interconnected with cytoskeleton, are also critical for cell signal propagation [135,136], but they are outside of the scope of this review. 

### 4.1. S1P Signaling in Mitochondria

Mitochondria form a highly dynamic network throughout the cytoplasm. The cells continuously adjust the rate of mitochondrial fission and fusion in response to changes in energy demands [132]. GTPases are key components involved in regulating fission and fusion. In particular, dynamin-related protein 1 (Drp1) is required for mitochondrial division from yeast to mammals; mitofusin 1 (MFN1) and mitofusin 2 (MFN2) mediate mitochondrial outer membrane fusion in vertebrates, while the dynamin-related GTPase Optic atrophy 1 (Opa1) is required for inner membrane fusion. MFN2 is also involved in other cellular functions, including mitochondria-endoplasmic reticulum contact site formation and stability, mitochondria-lipid droplet interactions, cellular proliferation, metabolic signaling, and mitophagy [137]. 

Imbalanced mitochondrial dynamics is observed in a range of diseases including cardiac [138] and neurodegenerative disorders [134]. Indeed, mutations that disrupt Opa1 and MFN2 cause autosomal dominant optic atrophy and axonal Charcot–Marie–Tooth disease type 2A (CMT2A) [139,140]. Recent studies suggest that skeletal muscle contraction might modulate both mitochondrial biogenesis and dynamics in a coordinated manner. High-intensity aerobic exercise has been reported to induce the protein expression of MFN1 and Fis1 [141,142,143,144] and a single bout of aerobic exercise increases the messenger RNA expression of MFN1 and MFN2 in the skeletal muscle of rats [138] and humans [145]. Since exercise-induced expression of PGC-1α precedes the induction of MFN1 and MFN2 [145], it has been suggested that mitochondrial biogenesis as well as mitochondrial fusion and fission may be regulated by PGC-1α [139]. It is interesting to note that induction of the canonical PGC-1α isoform in skeletal muscle promotes mitochondrial biogenesis and an oxidative phenotype, which mediates adaptive responses to endurance exercise [146]. In contrast, resistance exercise activates the expression of PGC-1α2, - 1α3, and - 1α4 [147]. 

Interestingly, S1P signaling is linked to mitochondrial biology. S1P modulates mitochondrial respiration and antagonizes the opening of the mitochondrial permeability transition pore in cardiomyocytes [56,148,149,150] and in HeLa cells [151]. S1P directly interacts with PHB2, which is a highly conserved chaperone that regulates the mitochondrial respiratory complex assembly and activity [56]. S1P modulation of mitochondrial function in pancreatic β-cells is associated with regulation of the expression of key regulators of mitochondrial dynamics, such as OPA1, MFN1, and PHB2 [152]. Recently, it has been reported that SphK1 regulates MFN2 fragmentation resulting in increased mitochondrial Ca^2+^ and downstream cellular effects. Indeed, SphK1 overexpression enhances cellular respiration and cell migration [151]. In addition, mitochondrial SphK1 is able to activate the mitochondrial unfolded protein response (mtUPR) in *C*. *elegans* [153]. External or internal stimuli may induce cellular stress responses, a well-orchestrated process finalized to cellular homeostasis recovery or to cell death. These responses include the endoplasmic reticulum stress response and mtUPR [154]. Although it is beyond of the scope of these review, a relevant role of SLs has been reported in endoplasmic reticulum stress response involved in many metabolic and neurodegenerative diseases [155,156,157,158]. mtUPR is evolutionarily conserved and triggers a retrograde signaling pathway from mitochondria to the nucleus leading to the upregulation of chaperones in order to maintain mitochondrial protein homeostasis, when mitochondria have suffered perturbations in protein folding in oxidative phosphorylation or DNA damage. Defects in mtUPR are involved in several diseases and aging. Indeed, pharmacological enhancement of mtUPR with small-molecule agents ameliorates mitochondrial and contractile dysfunction in the overloaded murine heart [159]. Whether SphK and mtUPR are interconnected in skeletal muscle pathology deserves to be investigated.

Recent studies have highlighted the potential molecular crosstalk among mitophagy, autophagy, and S1P signaling [160,161]. PHB2, the mitochondrial protein which binds S1P (see above), is also a receptor for mitophagy, which associates with the autophagosomal membrane-associated protein LC3 upon mitochondrial depolarization and outer membrane rupture [162]. SphK2, can increase autophagy independently of its catalytic activity since it is able to directly interact with Bcl-2 and displace Beclin-1 [163]. S1P metabolism is also involved in the LC3 lipidation [164]. In fact, S1PL catalyzes the hydrolysis of S1P to form hexadecenal and ethanolamine phosphate, which can be directed to the synthesis of phosphatidylethanolamine, the phospholipid that anchors LC3-I to phagophore membranes in the form of LC3-II. In addition, it has been recently reported that pharmacological inhibition of SphK1 or its knock-down by means of shRNA significantly reduces the expression of the regulators of mitophagy Bnip3l/Nix and Pink1 (PTEN-induced putative kinase 1) in a dose-dependent manner, thus modulating terminal erythroid differentiation [165].

### 4.2. S1P Signaling in the Nucleus

Nuclear matrix contains glycophospholipids, SLs, and cholesterol. In particular, SLs include SM, the most abundant nuclear SL, Cer, C1P, Sph, S1P, and gangliosides [166,167]. A recent study on intra-nuclear SM localization, obtained by using neutral sphingomyelinase conjugated to colloidal gold particles, revealed that SM is preferentially localized within the peri-chromatin region [168]. Moreover, Scassellati et al. (2010) [168] suggested the presence of SM at the nucleolus. SM has also been found associated with RNA in a complex that affords protection from RNAses. The presence of SM and the enzymes responsible for its metabolism in nuclear membranes as well as in the nuclear matrix may suggest that the signaling starting from SM degradation contributes to regulate not only nuclear membrane properties, such as those correlated to membrane fluidity, but also intranuclear functions. Moreover, other enzymes necessary for the autonomous metabolism of Cer and Sph are present into the nucleus [161], opening interesting perspectives for SLs peculiar nuclear functions, such as DNA duplication and regulation of chromatin remodeling and gene expression/transcription (reviewed in [169,170]).

SphK2/S1P signaling axis in the nucleus and its role in gene regulation via epigenetic mechanism(s) is an emerging field of the research. In the nucleus, S1P is less abundant compared to Sph and Cer, but, as reported in several studies, its level is strictly regulated. For instance, S1P content increases after infection with *P. aeruginosa* in epithelial cells and in embryonic fibroblasts obtained from S1PL1 knockout mice (sgpl1 ^−/−^) [171]. An accumulation of the bioactive lipid also occurs after inhibition of S1PL by THI (2-acetyl-5- tetrahydroxybutl imidazole) [172]. Increased nuclear S1P level is reported in MCF-7 cells overexpressing SphK2, whereas reduced level occurs in down-regulated SphK2 cells [59]. Similarly, pharmacological inhibition of SphK2 activity with its rather specific inhibitor, ABC294640, reduces the amount of S1P in the nucleus [173]. In addition, SphK2 can promote the phosphorylation of fingolimod (FTY720) and its accumulation either in the cytoplasm or in the nuclei of mouse embryonic fibroblasts, SH-SY7Y neuroblastoma, and hippocampal cells [174,175]. Notably, the localization and the expression of selective S1PRs has been correlated with tumor progression and prognosis. In fact, elevated level of cytoplasmic S1PR1 and nuclear S1PR2/S1PR3 expression are associated with breast cancer survival [176]. Moreover, a role for S1PR4 in breast cancer prognosis has been reported [65]. Finally, it has been documented that, in mammalian cells, S1PR5 co-localizes with SphK1/2 in the centrosome, a site for active GTP-GDP cycling involving G(i) and tubulin, which are required for spindle pole organization [64]. This finding may suggest a function for S1PR5 as a ligand-activated guanine nucleotide exchanger factor (GEF) in regulating cellular division. Despite these reports, further studies are necessary to identify the exact role of elevated nuclear S1P or S1P analogue signaling in the appearance and progression of diseases. Hait and collegues (2009) [59] reported for the first time that S1P can associate with histone H3 and control the binding of acetyl groups to lysine residues within histone tails through HDAC1/HDAC2 regulation [59,175]. In particular, nuclear SphK2/phospho-SphK2 and S1P co-immunoprecipitated with HDAC1/2 leading to increased histone acetylation at several lysine residues of histone H3, as reported in MCF-7 and MLE-12 cells, and mouse embryonic fibroblasts [72,173,174,175]. Therefore, HDACs are direct intracellular targets of SphK2/S1P axis and SphK2 is a part of repressor complex and support the involvement of S1P in the epigenetic regulation of gene expression. In general, in mammalian cells, the chromatin remodeling is the result of two major co-repressor complexes Sin3a, and NuRD/Mi2 [177,178]. It has been demonstrated that SphK2 and the active phospho-SphK2 associate with HDAC1/2 as part of these two co-repressor complexes. The formation of these complexes at the promoter of genes encoding pro-inflammatory cytokines [173], is crucial for the ability of SphK2 to modulate inflammation and tissue injury. For instance, it has been reported that genetic deletion of SphK2, but not of SphK1, protected mouse against *P. aeruginosa* mediated lung inflammation and pro-inflammatory cytokines secretion (i.e., IL-6 and TNF-α) [179]. Moreover, in epithelial cells SphK2/S1P activation is able to control the cytokine secretion through HDAC1/2 [179]. Although it has been shown that DHAC inhibitors (i.e., trichostatin A) and unlabeled S1P compete with [^32^P] labelled S1P [59], the exact molecular mechanism(s) of S1P-mediated inhibition of HDAC1/2 activity and the significance of this regulation in vivo remains to be carefully investigated.

## 5. S1P/S1PR Signaling in Neuromuscular Disorders

Most of the neurodegenerative diseases, which affect millions of people worldwide, shares alterations of mitochondria structure/function as common features.

### 5.1. Charcot-Marie-Tooth Disease

CMT disease, the most commonly inherited neurologic disorder, is characterized by motor and sensory neuropathy gait abnormalities, hammer toes, diminished sensation, and deep tendon reflexes [180,181]. The disease shows all modes of inheritance (autosomal dominant, autosomal recessive, X-linked). Autosomal dominant inheritance is the most frequent, however mutations resulting in sporadic cases have been identified.

CMT is classified on the base of mutations and clinical associations [180]:

CMT1 (caused by a duplication of the *PMP22*, the gene encoding peripheral myelin protein 22 kD), characterized by reduced motor nerve conduction velocity, evidence of demyelination and remyelination, is present in almost 80% of cases;

CMT2 (caused by mutations in MFN2) shows normal or slightly reduced nerve conduction velocity, and decreased muscle action potential, alterations in the neuromuscular relations, demyelination and axonal degeneration;

CMTX1 (caused by a mutation in *GJB1* (gap junction protein beta 1), a gene that encodes the gap junction protein connexin 32) displays intermediate motor conduction velocities and stroke-like episodes and white matter modifications;

CMTXB (caused by mutations of *MPZ* (myelin protein zero), which is the major constituent of peripheral myelin) exhibits reduced motor nerve conduction velocities, demyelination; CMT3, has infantile (Dejerine-Sottas) or birth onset and is a hypomyelinating neuropathy; CMT4 autosomal recessive is a demyelinating subtype present in less than 10% of the european CMT population.

Several evidence indicate the involvement of SLs and, in particular, S1P and Cer in CMT, emphasizing the importance of sphingoid molecules for neuromuscular functions in this disease [182]. Plasma S1P level is increased in some CMT patients with S1PL deficiency that seems the cause of a distinct form of CMT disease in humans. Indeed, heterozygous mutations of *SGPL1* gene cosegregate with disease: p.Ser361* mutation triggers nonsense-mediated mRNA decay and the missense p.Ile184Thr mutation causes partial protein degradation. In the same report it has also been demonstrated that the downregulation of the S1PL orthologue in neurons impairs the morphology of neuromuscular junction in *Drosophila,* which shows a reduction in branching and in the number of synaptic boutons [182].

Schwartz et al. (2017) [183] have demonstrated that mutations in heat shock protein 27 (Hsp27) in CMT2F patients are responsible for the decrease of Cer level in mitochondria, leading to larger and interconnected organelles with reduced respiratory function, increased autophagy and, in turn, to neuronal degeneration. In another cell system, neonatal rat cardiomyocytes, Cer stimulates fragmentation of the mitochondrial network and activation of cardiomyocyte apoptosis, which is accentuated by decreasing MFN2 expression [184] confirming the correlation among Cer, MFN2, and mitochondrial dynamics. Further investigation is needed to define whether CMT2F dysfunction is due to Cer decrease or impairment of interconnected sphingoid molecules such as S1P into the mitochondria. Collectively, these reports underline an important crosstalk between SLs and mitochondrial structure/function at two distinct levels: the mitophagy and the quality-control mechanisms mediated by heat-shock proteins necessary for preventing the accumulation of misfolded proteins.

### 5.2. Myasthenia Gravis

Myasthenia gravis (MG) is an autoimmune neurological disease that is characterized by the expression of anti-acetylcholine receptor (AChR) antibodies. The immune response at AChRs of neuromuscular junction leads to skeletal muscle fatigue, which is aggravated following periods of activity and alleviated following rest [185]. Treatment for MG includes acetylcholinesterase inhibitors, immunosuppressive medications, plasma exchange, intravenous immuglobulis, and, in some cases, thymectomy [185].

Fingolimod (FTY720) has been approved by the Federal Drug Amministration and also by several European countries for the treatment of multiple sclerosis, while the results of phase III trial for siponimod have been recently published [38]. Siponimod, another sphingosine analogue, is able to reduce the risk of disability progression in secondary progressive multiple sclerosis [38]. These compounds are being tested in several neurodegenerative and neuromuscular diseases [29]. Recently, siponimod, and fingolimod (FTY720), that can be converted to S1P analogues inside the cells (see Section 3.1), have been tested in animal models of MG. Experimental autoimmune MG has been established by immunizing mice with AChR from *Torpedo californica* [186,187] or with the immunogenic region 97–116 of the rat AChR α-subunit [188]. Fingolimod (FTY720) prevents MG induction in mice [181] and ameliorates the symptoms and reduces inflammatory cytokines in rats [183]. However, fingolimod (FTY720) or siponimod when administered to mice after the onset of the disease, inhibits lymphocyte egress, resulting in peripheral lymphopoenia without changing antibody titer, total or antigen-specific plasma cell populations and the severity of the disease [187]. The authors have suggested that differences in the mechanisms of MG, an autoantibody-mediated disease, and multiple sclerosis, a rather T-cell-mediated disease, might explain the different outcome of the drugs in these pathologies.

### 5.3. Duchenne Muscular Dystrophy

Duchenne muscular dystrophy is a lethal muscle-wasting disease affecting 1:3500 males. It leads to muscle degeneration and death due to cardiorespiratory complications. Duchenne muscular dystrophy originates from mutations in the dystrophin gene [189]. The dystrophin protein is part of the dystrophin-glycoprotein complex that has a crucial role in the integrity and connectivity of muscle plasma membrane with the extracellular matrix and the muscle cytoskeleton.

Studies in Drosophila, which does not have identified S1PRs, have shown that genetic increase of the levels of the S1P or delivery of the S1P lyase inhibitor THI suppresses dystrophic muscle degeneration [190]. In mice S1P can act as a ligand for S1PRs and as a HDAC inhibitor. In mdx mice, a model for Duchenne muscular dystrophy exhibiting a point mutation of the dystrophin gene, muscular S1PL expression is higher compared to control muscles, while S1P plasma level is reduced [4] Upregulation of S1P by THI increases regeneration and muscle force in mdx mice [4,5]. The beneficial effects of THI correlate with significantly increased nuclear S1P, decreased HDAC activity and increased acetylation of specific histone residues which promote the up-regulation of skeletal muscle metabolic genes, such as micro RNAs miR-29 and miR-1 [172].

The experiments in mdx mice performed by Loh et al. [4] have suggested that the improvement of the regenerative ability following S1PL inhibition depends on S1PR2 through suppression of Rac1, activation of STAT3, repression of cell cycle inhibitors and entry of the satellite cells into the cell cycle. Other studies suggest that decrease of S1PR3 might improve the phenotype of mdx mice. As already stated in the introduction, overexpression of S1PR3 suppresses cell cycle in satellite cells, while skeletal muscle regeneration is enhanced in S1PR3-null mice [14]. These authors have investigated the effect of the knock-out of this receptor subtype on mdx mice and have found an increase in the size of myofibers of different muscles in mdx/S1PR3 ^−/−^ compared to mdx/SP1R3 ^+/−^.

More recently, it has been reported that S1PR3 is a target of miR-127 both in vivo and in vitro [15]. miR-127 accelerates regeneration by promoting satellite cell differentiation, but S1PR3 overexpression eliminates the miR-127- mediated increase of skeletal muscle cell differentiation.

Another devastating dystrophic disease is the limb girdle muscular dystrophy type 2C, which is caused by mutations in δ-sarcoglycan. A recent study in a mouse model of this pathology has shown that fingolimod (FTY720) is an effective treatment when therapy is initiated early. Although the mechanism was not investigated, fingolimod (FTY720) decreased fibrosis and elevated protein levels of δ-sarcoglycan, a dystrophin-glycoprotein family member [191].

## 6. Conclusions

In conclusion, the preliminary studies reviewed above underline that S1P/S1PR signaling may be crucial for the development or/and the progression of neuromuscular diseases (Figure 2). Therefore, the involvement of a distinct crosstalk between S1P/S1PR signaling and specifically localized effectors is worthy of being further investigated. In particular, the novel findings regarding SLs and CMT open important questions on how increased S1P level alters the interconnection between sphingoid molecules, such as the Cer/S1P ratio, at the different subcellular locations (plasma membrane, mitochondrial, nuclear, or endoplasmic reticulum) and in different cell types (neuron, muscle fiber, glia, neural, and satellite cells). Moreover, a deeper understanding of S1P/S1PR signaling in mitochondrial dynamics, excitability, and epigenetic modulation of gene expression in neural/muscular stem cell proliferation and differentiation is also a crucial and urgent point to be investigated. Controversial results have been obtained with SL analogues in MG. In fact, fingolimod (FTY720) treatment is not effective when supplied after the immunization in mice models, but some improvement has been found in the rat model.

Therefore, further detailed studies are needed for the translation of fingolimod (FTY720) effects to the clinic. Modification of S1P levels and signaling by using miRNAs might also be a therapeutical tool in neuromuscular diseases. Indeed, the overexpression of miR-127, which targets S1P3R, reduces some pathological features in mdx mice skeletal muscle, extends the treadmill running time, and increases muscular force [15]. Specific SPL inhibitors and/or S1P precursors or SphK activators, as well as antagonist of S1P3R and agonists of S1P2R may be useful in Duchenne muscular dystrophy. The translation of sphingolipid basic research to therapeutic opportunities in skeletal muscle diseases is still very preliminary, but new-generation drugs developed to more specifically target S1PR—as well as many of the enzymes involved in S1P metabolism—will hopefully open new therapeutic opportunities also in this field.

## Figures and Tables

**Figure 1 ijms-20-06364-f001:**
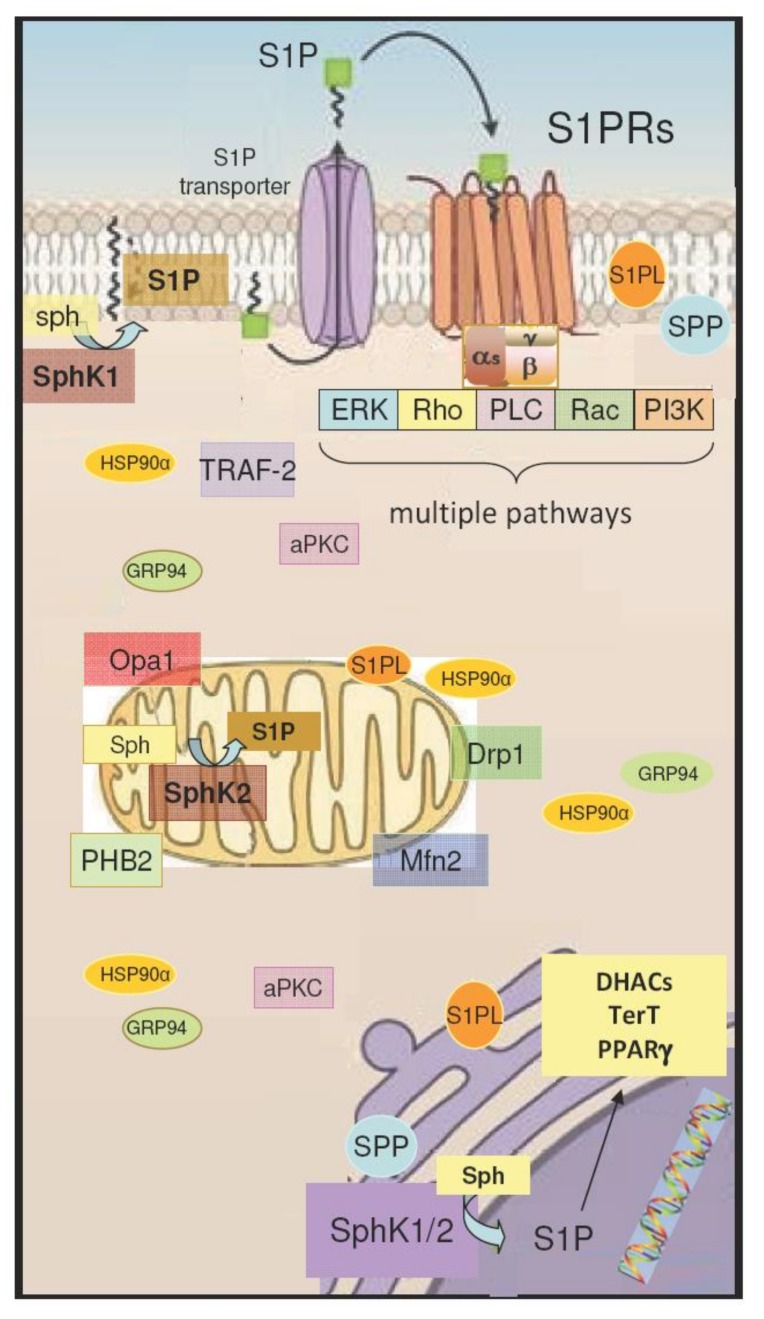
S1P receptor-activated pathways and intracellular targets of S1P. S1P is synthesized from sphingosine (Sph) by the sphingosine kinase isoforms SphK1 and SphK2, and it is irreversibly cleaved by S1P lyase (SPL), or dephosphorylated by S1P phosphatases (SPP) localized mainly in the endoplasmic reticulum and also in the nucleus. S1P produced inside the cell can be transported to the intercellular space by an S1P transporter. As a ligand, S1P acts as autocrine and paracrine factor triggering specific signaling pathways by interacting with S1P specific heterotrimeric GTP binding protein-coupled receptors (S1PRs). S1PR activation modulates extracellular signal–regulated kinases (ERK), Rho and Rac GTPases, phospholipase C (PLC), and phosphoinositide 3-kinases (PI3K) and, in turn, multiple signaling pathways. Subcellular localization of S1P intracellular targets is indicated: cytoplasm for atypical protein kinase C (aPKCs), tumor necrosis factor receptor-associated factor 2 (TRAF-2), mitochondria for prohibitin 2 (PHB2), dynamin-related protein 1 (Drp1), mitofusin 2 (Mfn2), optic atrophy 1 (Opa1), nucleus for histone deacetylases (HDACs), telomerase (TerT), and poly (ADP-ribose) polymerases (PARP). Moreover, S1P specifically interacts with the N-terminal domain of the heat shock proteins GRP94 and HSP90αγ.

**Figure 2 ijms-20-06364-f002:**
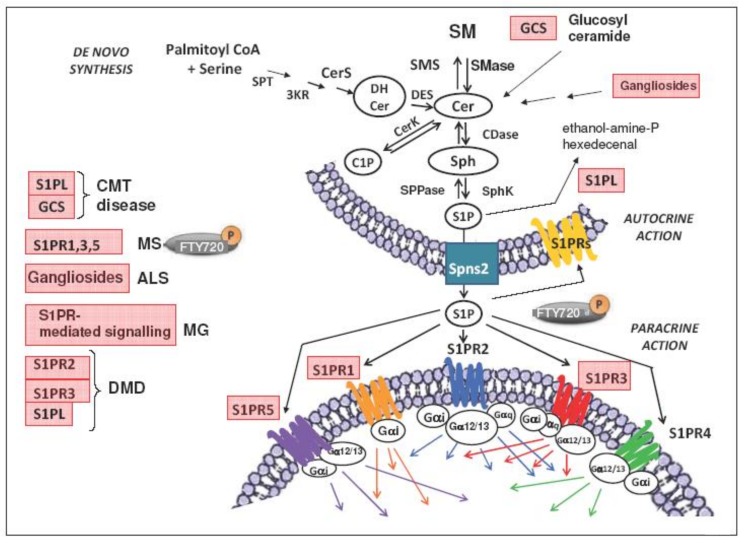
Sphingolipid metabolism and S1P/S1P receptors signaling and summary of the metabolic pathways associated with neuromuscular diseases. Disease names in light red are linked to defects in the sphingolipid metabolism /S1P/S1PR signaling. Ceramide (Cer) is formed via the sphingomyelin (SM) cycle or de novo sphingolipid synthesis involving serine palmitoyl transferase (SPT), 3-keto reductase (3KR), ceramide synthase (CerS), and desaturase (DES), and converted reversibly to sphingosine (Sph) by ceramidase (CDase), or phosphorylated to ceramide-1-phosphate (C1P) by ceramide kinase (CerK) activity. S1P is synthesized from sphingosine (Sph) by the sphingosine kinases and irreversible cleaved by S1P lyase (S1PL), which generates hexadecenal and phosphoethanolamine. S1P is also a substrate of specific S1P phosphatases (SPPase). S1P produced inside the cell can be transported in the intercellular space by an ATP-binding cassette transporter named spinster homolog 2 (Spns2). As a ligand, S1P acts as autocrine and paracrine factor triggering specific signaling pathways by interacting with S1P specific heterotrimeric GTP binding protein-coupled receptors, named S1PR. After ligand binding, the G protein α subunit (Gαs, Gαi, Gαq, Gα12/13), together with the bound GTP, dissociates from the β γ complex and they affect intracellular signaling proteins or target functional proteins directly. Five subtypes of S1PRs have been described and appear to be selectively expressed in skeletal muscle cells and the nervous system. Other abbreviations: SMase, sphingomyelinase; ALS, amyotrophic lateral sclerosis; CMT, Charcot–Marie–Tooth disease; DMD, Duchenne muscular dystrophy; GCS, glucosylceramide synthase; MG, myastenia gravis; MS, multiple sclerosis [29]; SMS, sphingomyelin synthase; SPT, serine palmitoyltransferase. The signalling pathways downstream to each specific S1P receptor subtype are drawn as arrows of the same color of the S1P receptor

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
