# Peer review of "S1P/S1P Receptor Signaling in Neuromuscolar Disorders"

_ijms, 2019, doi:10.3390/ijms20246364_

Round 1

Reviewer 1 Report

The manuscript from Dr. Meacci and Dr. Garcia-Gil entitled “S1P/S1P receptor signalling in neuromuscular disorders” is a comprehensive and excellently written review manuscript. The topic is well reflected and only some minor comments should be addressed.

The authors reflect on both the role of the S1P/S1PR system in neurological diseases such as multiple sclerosis and in the nervous system in general as well as on its role in certain cancer types. One aspect that is missing however, is the recently described role of S1P and its receptors also in neurological tumours, i.e. malignant gliomas. Perhaps, a short paragraph on this topic may be added to the manuscript to complete the picture of the complex actions of S1P in the brain.

While the title of the manuscript suggests that the authors would focus on the topic of “S1P/S1PR signalling in neuromuscular disorders” the very same chapter of the actual manuscript text is quite short (only about two pages) compared to the rest of the manuscript (about 8 pages). To adjust this imbalance a little more towards the actual topic of the review, the authors could include a table or - if they think this would be suitable – an additional figure summarizing the particular role of S1P in neuromuscular disorders.

At the end of the manuscript (in the conclusion section), the authors point towards the controversial results which have been obtained with SL analogues in MG. In addition, they mention miR-127 as particular target to modify SL signalling as a therapeutic tool for dystrophy in humans. This conclusion is highly speculative and again does not really fit with the tenor of the manuscript which should focus on S1P. Here, a broader outlook i.e. for the use of compounds specifically modifying S1P signaling would be desirable.

Author Response

Dear Editor,

First of all we want to thank you and the reviewers for the thorough evaluation of our manuscript entitled “S1P/S1P receptor signalling in neuromuscolar disorders" by Meacci E and Garcia-Gil M and for encouraging resubmission after appropriate corrections. Accordingly, we revised the text adding a new figure (Figure 2) and extending some concepts in the text. We believe that the quality of our revised manuscript has been increased substantially, and hope that it can now be acceptable for publication.

Reviewer 1.

Schematic representation of sphingolipid metabolism might help to picture its complexity. It might also point the major targets in neuromuscular diseases.

In the new version of the review a schematic representation of sphingolipid metabolism and summary of the metabolic pathways associated with neuromuscular diseases, in which the disease names (light red) are linked to defects in the sphingolipid metabolism and S1PR signalling is now included.

The described paradigm of “inside-out” signaling that involved flip-flopping of S1P through the plasma membrane required more explanation, since the removal of plasma membrane phospholipid required the high energy costs.

The paragraph has been rewritten and a new reference (new 69) has been added (Reitsema, V. ; Bouma, Hjalmar ; Kok, Jan. Sphingosine-1-phosphate transport and its role in immunology. AIMS Molecular Science 2014, 1, 183-201; DOI: 10.3934/molsci.2014.4.183).

Immunomodulatory ability of FTY720P might be mentioned in paragraph 3.1, since that described effect on CNS might involved peripheral action of this molecules Information on some clinical trials underline the potential of FTY720 treatment in neuromuscular disorder might be included

The immunomodulatory action of FTY720P and information on clinical trials of FTY720 in sclerosis multiple and amyotrophic lateral sclerosis as well as of another S1P analogue in multiple sclerosis have now been included in section 3.1.

New references 121-124 have been added:

Cohen, J.A. et al., Extended treatment with fingolimod for relapsing multiple sclerosis: the 14-year LONGTERMS study results. Ther. Adv. Neurol. Disord. 2019, 12,1756286419878324; 

Berry, J.D.; et al.,  Phase IIa trial of fingolimod for amyotrophic lateral sclerosis demonstrates acceptable acute safety and tolerability. Muscle Nerve 2017 56(6),1077-1084;  DOI: 10.1002/mus.25733.

Koscielny ,V. Phase III SUNBEAM and RADIANCE PART B trials for Ozanimod in relapsing multiple sclerosis demonstrate superiority versus interferon-β-1a (Avonex®) in reducing annualized relapse rates and MRI brain lesions. Neurodegener. Dis. Manag. 2018, 8(3),141-142; DOI: 10.2217/nmt-2018-0012.

Cohen, J.A. et al., RADIANCE Trial Investigators. Safety and efficacy of ozanimod versus interferon beta-1a in relapsing multiple sclerosis (RADIANCE): a multicentre, randomised, 24-month, phase 3 trial. Lancet Neurol. 2019 18(11),1021-1033; DOI: 10.1016/S1474-4422(19)30238-8.

Reviewer 2 Report

This review summarized current understanding of how the localized actions of S1P and/or S1PR 66 signaling affect different biological processes associated with physiology and pathophysiology of muscle and neuronal tissues. The Authors focus on the role of S1P/S1PR signaling in the control of mitochondrial and nuclear functions, that are much more complex than previously thought. Part of this review is also describing the involvement of S1P and S1PR-mediated signaling as potential non-protein targets in neuromuscular diseases including Charcot-Marie-Tooth disease (CMT), myasthenia gravis (MG) and Duchenne muscular dystrophy.

Schematic representation of sphingolipid metabolism might help to picture its complexity. It might also point the major targets in neuromuscular diseases. The described paradigm of “inside-out” signaling that involved flip-flopping of S1P through the plasma membrane required more explanation, since the removal of plasma membrane phospholipid required the high energy costs. Immunomodulatory ability of FTY720P might be mentioned in paragraph 3.1, since that described effect on CNS might involved peripheral action of this molecules Information on some clinical trials underline the potential of FTY720 treatment in neuromuscular disorder might be included.

Author Response

Dear Editor,

First of all we want to thank you and the reviewers for the thorough evaluation of our manuscript entitled “S1P/S1P receptor signalling in neuromuscolar disorders" by Meacci E and Garcia-Gil M and for encouraging resubmission after appropriate corrections. Accordingly, we revised the text adding a new figure (Figure 2) and extending some concepts in the text. We believe that the quality of our revised manuscript has been increased substantially, and hope that it can now be acceptable for publication.

Reviewer 2.

The manuscript from Dr. Meacci and Dr. Garcia-Gil entitled “S1P/S1P receptor signalling in neuromuscular disorders” is a comprehensive and excellently written review manuscript. The topic is well reflected and only some minor comments should be addressed.The authors reflect on both the role of the S1P/S1PR system in neurological diseases such as multiple sclerosis and in the nervous system in general as well as on its role in certain cancer types.

One aspect that is missing however, is the recently described role of S1P and its receptors also in neurological tumours, i.e. malignant gliomas. Perhaps, a short paragraph on this topic may be added to the manuscript to complete the picture of the complex actions of S1P in the brain.

 In this minireview, our intention was to give an update on the specific topic of the potential role of S1P/S1P receptor signalling in the appearance and progression of  neuromuscolar disorders  focusing especially on cell degeneration. The role of S1P on the proliferation of precursor cells has been mainly  mentioned taking into consideration that this is a crucial biological process for tissue repair and survival. The role of S1P and its receptors in glioblastomas is complex, probably due to their heterogeneity. Therefore , we feel that it is really hard to resume all the concepts in a short paragraph.

To adjust this imbalance a little more towards the actual topic of the review, the authors could include a table or - if they think this would be suitable – an additional figure summarizing the particular role of S1P in neuromuscular disorders.

In the new version of the review we have included a schematic representation of sphingolipid metabolism and summary of the pathways associated with neuromuscular diseases. The disease names (light red) linked to defects in the sphingolipid metabolism and S1PR signalling are now included. Moreover, actually no many data regarding S1PR-mediated signalling are available.

6      At the end of the manuscript (in the conclusion section), the authors point towards the controversial results which have been obtained with SL analogues in MG. In addition, they mention miR-127 as particular target to modify SL signalling as a therapeutic tool for dystrophy in humans. This conclusion is highly speculative and again does not really fit with the tenor of the manuscript which should focus on S1P. Here, a broader outlook i.e. for the use of compounds specifically modifying S1P signaling would be desirable.

The conclusion section has been rewritten